# Taxonomic identification and temperature stress tolerance mechanisms of Aequorivita marisscotiae sp. nov

Wenqi Liu[1,2], Bailin Cong [ID] [1,2 ✉], Jing Lin[1], Shenghao Liu[1], Aifang Deng[1] & Linlin Zhao[1]

The deep sea harbours microorganisms with unique life characteristics and activities due to adaptation to particular environmental conditions, but the limited sample collection and pure culture techniques available constrain the study of deep-sea microorganisms. In this study, strain Ant34-E75 was isolated from Antarctic deep-sea sediment samples and showed the highest 16 S rRNA gene sequence similarity (97.18%) with the strain *Aequorivita viscosa* 8-1b[T]. Strain Ant34-E75 is psychrotrophic and can effectively increase the cold tolerance of *Chlamydomonas reinhardtii* (a model organism). Subsequent transcriptome analysis revealed multiple mechanisms involved in the Ant34-E75 response to temperature stress, and weighted gene co-expression network analysis (WGCNA) showed that the peptidoglycan synthesis pathway was the key component. Overall, this study provides insights into the characteristics of a deep-sea microorganism and elucidates mechanisms of temperature adaptation at the molecular level.

[1] First Institute of Oceanography, Ministry of Natural Resources, Qingdao 266061, China. [2] These authors contributed equally: Wenqi Liu, Bailin Cong.
✉email: biolin@fio.org.cn

The extreme conditions in the deep sea, including a lack of light and oxygen, temperatures ranging from 0 to 400 °C, and high hydrostatic pressure, make it challenging for life to exist in these habitats[1]. To adapt to these extreme conditions, organisms in the deep sea have evolved various adaptation and cell signaling mechanisms[2]. Nonetheless, the deep ocean remains poorly studied due to difficulties in sampling and culturing its microbiota[3]. There are a large number of rare species in the deep-sea environment, where more than half of species are new to science, and more than 95% of the species in some taxa are undescribed[3]. In this study, a new species of the genus *Aequorivita* was identified from deep-sea sediment at a depth of 2560 m in the Scotia Sea.

The genus *Aequorivita*, was first described by Bowman & Nichols in 2002 and belongs to the family *Flavobacteriaceae*, within the order *Flavobacteriales*[4], and it has since been emended by Park[5]. Additionally, species of the genus *Vitellibacter* were transferred to the genus *Aequorivita* in 2016[6]. *Aequorivita* comprises aerobic, Gram-negative, non-spore-forming bacteria, and its major fatty acids are anteiso-$C_{15:0}$, iso-$C_{15:0}$ and iso-$C_{17:0}$3-OH, with menaquinone-6 (MK-6) as the major respiratory quinone. The genus currently comprises 16 species with valid published names (https://lpsn.dsmz.de/genus/aequorivita), which have been isolated from terrestrial and marine Antarctic habitats[4], samples of marine algae collected from the South Sea in the Republic of Korea[5], estuarine sediments of the Pearl River in China[7], and the intertidal zone and sediment of the East China Sea[8,9]. However, previous studies on *Aequorivita* have only addressed their taxonomy, fatty acid composition and DNA G + C content. In 2018, the first comprehensive study of the chemistry and biological activity of *Aequorivita* was conducted. An extract of *Aequorivita* sp. was found to possess antimicrobial and anthelmintic properties affecting multidrug-resistant bacteria and the nematode *Caenorhabditis elegans*[10]. Through a combination of LC–$MS^2$-based dereplication and antibacterial assay using the clinically relevant bacterial test strain *Staphylococcus aureus* (methillicin-resistant, DSM 18827), seven N-terminal glycine- or serine-bearing iso-fatty acid amides were isolated from *Aequorivita* sp., three of which were newly discovered[11]. To date, there are relatively few studies of this genus, but they have shown that this genus is worthy of study. Therefore, after the identification of strain Ant34-E75 as a new species of *Aequorivita*, we explored the function of the strain based on experimental evaluations in this study.

In this study, the experimental strain of *Aequorivita* was shown to be capable of growing between 5-37°C, with an optimum growth temperature of 27°C, and was identified as a psychrotroph. Moreover, we discovered that quite a few *Aequorivita* bacteria were isolated from algae; therefore, the experimental strain exhibited algal-bacterial symbiosis. *Chlamydomonas reinhardtii* was chosen for culture with this strain, since this species is often used as a model organism in various studies due to its short growth cycle, moderate cell size and clear genetic background[12–14]. The experimental results showed that the strain not only tolerated low temperatures but also increased the freezing tolerance of *Chlamydomonas reinhardtii*. The cold tolerance mechanism of the strain is a complex regulatory process, which includes as follows: (1) The adjustment of protein amino acid composition, such as the replacement of lysine to arginine in cold-adapted α-amylase from *Pseudoalteromonas haloplanktis*, can make its conformation more stable[15]; (2) High GC content is also more conducive to the stability of DNA strands[16]; (3) Regulating osmotic pressure, such as producing more extracellular carbohydrates and polymeric substances[17]; (4) The maintenance of the liquid crystalline state of the membrane, such as increasing the proportion of unsaturated fatty acids[18]; (5) The secretion of

cold-shock proteins, osmo-protection, membrane-related proteins and heat stress proteins, etc[19,20]. Consequently, the mechanisms of tolerance to temperature stress were investigated in this strain by transcriptome analysis.

In summary, we completed the taxonomic identification of a new strain, Ant34-E75, obtained from Antarctic deep-sea sediment, and proposed that it belongs to the genus *Aequorivita*, with the name *Aequorivita marisscotiae* sp. nov. with Ant34-E75 as the type strain. This work increases the understanding of deep-sea microorganisms through a systematic description of their characteristics. Transcriptomic analysis revealed the mechanisms of the temperature stress response in the strain, such as peptidoglycan synthesis, the ABC transport system, two-component system, amino acid composition adjustment in proteins, nitrogen metabolism, oxidative phosphorylation, and nicotinate and nicotinamide metabolism. Weighted gene co-expression network analysis (WGCNA) identified the peptidoglycan synthesis pathway as a key module related to temperature stress. Thus, we provide insights into the mechanisms of microorganismal adaptation to deep-sea environments.

## Results

**Polyphasic taxonomy of the strain**. According to polyphasic taxonomy, the properties of strain Ant34-E75 were generally consistent with the description of *Aequorivita*, including the morphological characteristics, respiratory quinones, major fatty acids, and G + C content of the whole genome of the strain[4]. Furthermore, the results showed that Ant34-E75 clustered within *Aequorivita* in the phylogenetic tree (Fig. 1a). However, compared with other strains of the same genus, there were many differences in enzymatic properties and other physical and chemical properties (Tables 1 and 2). The 16 S rRNA sequencing results showed that strain Ant34-E75 (accession number for 16 S rRNA is OR529246, and length of nucleotides obtained are 1,430 bp) presented a high similarity of 97.18% to the reference strain *Aequorivita viscosa* 8-1b$^T$ (accession number of 16 S rRNA is NR109011.1) and showed less than 97.0% similarity with other species of the genus *Aequorivita*. Additionally, the maximum average nucleotide identity value (Supplementary Table 1) was 78.30%, which was far below the level indicative of relatedness at the species level (95%)[21]. Digital DNA-DNA hybridization (dDDH) was performed to determine the similarity of the genomes of strain Ant34-E75 and *Aequorivita* viscosa 8-1b$^T$. The results showed that the dDDH was 20.0%, which was lesser than 70%. Moreover, the strain Ant34-E75 was also classified as d__Bacteria; p__Bacteroidota; c__Bacteroidia; o__Flavobacteriales; f__Flavobacteriaceae; g__*Aequorivita*; s__ by Genome Taxonomy Database (GTDB). Based on phenotypic and genotypic characterization, strain Ant34-E75 represents a novel species of the genus *Aequorivita*, for which the name *Aequorivita marisscotiae* sp. nov. is proposed. The type strain is strain Ant34-E75.

**Description of *Aequorivita marisscotiae* sp. nov**. *Aequorivita marisscotiae* (ma.ris.sco´ti.ae. L. neut. n. mare, sea; N.L. gen. n. marisscotiae, of the Scotia Sea).

The strain is Gram-negative, oxidase-negative, catalase-positive, rod-shaped, and 0.2–0.4 × 0.8–1.6 μm in size (Fig. 1b). It grows optimally in the presence of 2–4% NaCl, at 27°C, in a pH range of 6–8. It hydrolyses gelatine, but not sodium alginate, casein, starch, Tween 80, urea or xylan. It is capable of hydrolysing arginine, urea, esculin, gelatine and 4-nitrophenyl-β-D-galactopyranoside (API 20NE). Positive for lipoesterase C8, α-galactosidase, β-galactosidase, β-glucuronidase, α-glucosidase, β-glucosidase, α-mannosidase and β-fucosidase are present (API

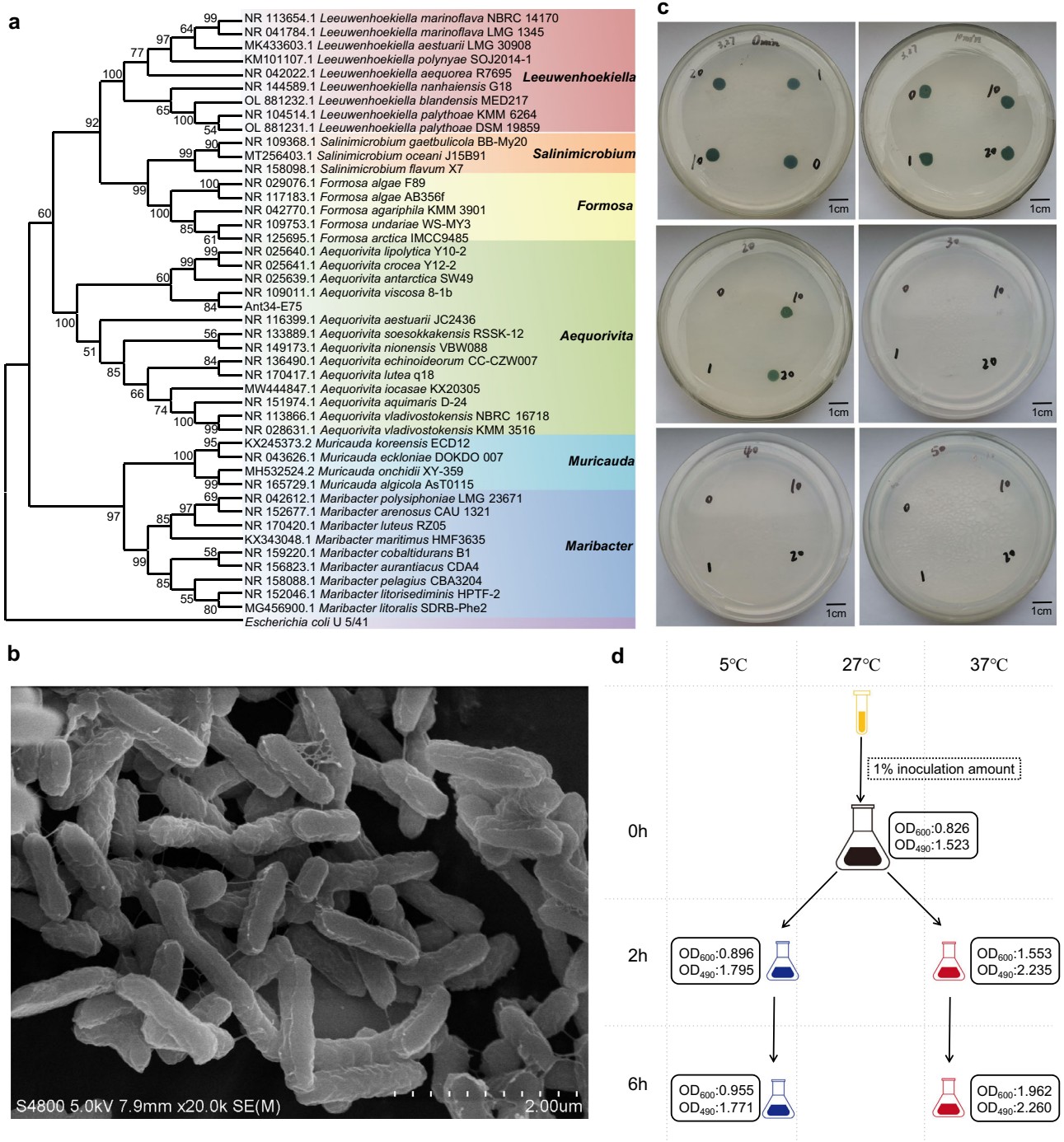

**Fig. 1 Taxonomic identification and experimental exploration of strain Ant34-E75. a** Maximum-likelihood phylogenetic tree produced by the comparison of the 16 S rDNA sequences. Numbers on the nodes are bootstrap values in percentages (1000 replicates). **b** Scanning electron microscopy image. **c** Algal-bacterial symbiosis. The number at the top of the plate is the time at which the coculture mixture was treated at –20 °C; the other number is the proportion of the bacteria and algae in the coculture mixture (volume ratio). **d** Treatment conditions of the sample for transcriptome sequencing. $OD_{600}$ is the bacterial concentration, and $OD_{490}$ is the polysaccharide content.

ZYM). Positive for pH 6, 1% NaCl, 4% NaCl, 1% sodium lactate, D-serine, rifamycin SV, gelatine, Glycyl-L-proline, L-aspartic acid, L-glutamic acid, lithium chloride, potassium tellurite, Tween 40, acetoacetic acid, acetic acid, aztreonam and sodium butyrate was observed in Biolog GENIII test system. The main respiratory quinone is menaquinone-6, and the major fatty acids are iso-$C_{15:1}$G, iso-$C_{15:0}$ and iso-$C_{17:0}$3-OH (Table 2). The polar lipid profile contains phosphatidylethanolamine, three unidentified aminolipids and one unidentified lipid.

The type strain is Ant34-E75 ( = CCTCC AB 2023079 = KCTC 92993), was isolated from deep-sea sediment collected from the Scotia Sea. The genome size of the type strain is 3,451,112 bp with a G + C content of 38.4 mol%, and the predicted number of coding genes was 3235, including 37 tRNA, 2 5 S rRNA, 2 16 S rRNA, 2 23 S rRNA and 1 sRNA. The complete genome sequence has been deposited in the NCBI database under accession number CP122379, and the GenBank accession number for the 16 S rRNA gene sequence was OR529246.

**Table 1 Comparison of the characteristics of strains belonging to the same genus.**

| Characteristic | 1 | 2 | 3 | 4 | 5 | 6 | 7 | 8 | 9 | 10 | 11 |
|---|---|---|---|---|---|---|---|---|---|---|---|
| NaCl range, optimum (%, w/v) | 0-8, 4-5 | 0.5-8, 1-3 | 0-8, 5 | 0.5-8, 1 | 1-10, 3-6 | 0-9, 2-6 | 0-4 | 1-6, 2 | 0.5-10 | 0.5-6 | 0-12, 2-3 |
| pH range, optimum | 5-9, 6-8 | 6.0-9.5, 7.0-8.5 | 5-8, 7 | 7-10, 9 | 6-11, 6-8 | 5-9, 7-8 | 6-9 | 5.5-10, 7.5-8.5 | | | 5.0-9.5, 7-8 |
| Temperature range, optimum(°C) | 5-37, 27 | 4-39, 28-32 | 10-40, 30 | 4-42, 30-35 | 4-41, 25-30 | 4-43, 37 | 20-40 | 4-43, 28 | | | 10-37, 30 |
| Catalase | + | + | + | + | + | + | + | + | | | |
| Oxidase | + | - | + | - | + | + | + | + | + | + | + |
| Gelatine | + | + | + | + | + | + | + | | + | + | + |
| Sodium alginate | - | - | | - | - | - | | - | - | - | - |
| Casein | - | - | - | + | + | + | | + | - | + | + |
| Starch | - | - | - | - | - | - | - | - | - | - | - |
| Tween 80 | - | + | | - | - | - | + | - | + | + | + |
| Tween 20 | - | + | | - | + | + | | + | + | + | + |
| Urea | - | + | | - | + | + | | - | - | | - |
| Xylan | - | - | | - | - | - | + | - | - | + | - |
| Respiratory quinone | MK-6 | MK-6 | MK-6 | MK-6 | MK-6 | MK-6 | MK-6 | MK-6 | MK-6 | MK-6 | MK-6 |
| Polar lipids* | PE, L1-3, AL1-3 | PE, AL1-3, GL1-4, L1-5 | PE, L1-2, GL1-2, PGL | PE, L1-2, AL1-2, GL, PGL1-3 | PE, AL1-2, L1-2 | PE, L1-2, GL1-2, PGL | PE, L1-2, AL1-2, APL | | PE, L1-5, AL1-2, GL1-5 | | PE, L1-7, AL, GL, PL |
| G + C (mol%) | 38.4 | 36.6 | 42.8 | 34.6 | 38.7 | 40.6 | 34.7 | 41.3 | 38.5 | 35.6 | 38.9 |

*PE phosphatidylethanolamine, L unidentified polar lipid, AL aminolipid, GL glycolipid, PGL phosphoglycolipid, APL aminophospholipid, PL phospholipids.
Strains: 1, Aequorivita mariscotiae Ant34-E75; 2, Aequorivita viscosa 8-1b[T]; 3, Aequorivita lutea q18[T]; 4, Aequorivita sinensis S1-10[T]; 5, Aequorivita iocasae KX20305[T72]; 6, Aequorivita aquimaris D-24[T73]; 7, Aequorivita echinoideorum CC-CZW007[T74]; 8, Aequorivita vladivostokensis KMM 3516[T75]; 9, Aequorivita antarctica SW49[T4,9]; 10, Aequorivita lipolytica Y10-2[T4]; 11, Aequorivita soesokkakensis RSSK-12[T76].

**Algal-bacterial symbiosis**. Based on an algal-bacterial symbiosis experiment, strain Ant34-E75 could effectively increase the freezing tolerance of *Chlamydomonas reinhardtii*. The results are shown in Fig. 1c. *Chlamydomonas reinhardtii* could grow after 10 min of cold treatment but could not grow if the treatment was increased to 20 min, when strain Ant34-E75 was inoculated at either 0% or 1%. The strain allowed *Chlamydomonas reinhardtii* to continue growing even after 20 minutes of cold treatment when it was inoculated at 10% or 20% and increased the cold tolerance time of the algae by 10 min after 10% or 20% inoculation. However, the algae did not grow when the cold treatment time was extended to 30 min (time points of 40 min and 50 min were tested). These results suggest that the newly isolated strain Ant34-E75 has the ability to increase the cold tolerance of *Chlamydomonas reinhardtii* and enhance the growth of the algae under cold stress.

**Transcriptome sequencing and differentially expressed genes (DEGs) analysis**. To explore the molecular mechanism of strain Ant34-E75 response to temperature stress, five groups of bioreactors were used for culture. The strain was cultured to the logarithmic growth phase at 27 °C and then cultured at different temperatures (5 °C for 2 h and 6 h and at 37 °C for 2 h and 6 h) (Fig. 1d). Transcriptome sequencing resulted in clean data with high quality (a minimum Q20 score of 98.82% and Q30 score of 95.31%) (Supplementary Table 2). The proportion of alignment for clean data mapped to the reference genome (the complete genome of the strain Ant34-E75) was 98.89 to 99.36% (Supplementary Table 3). The results of the differential gene expression analysis (Supplementary Data 1), which was performed using the parameters of a $|\log_2 FoldChange| > 1$ and $p$ value $< 0.05$, showed that there were more DEGs in the 6-h groups (538, 536) than in the 2-h groups (379, 272) (Fig. 2). The DEGs were annotated and classified into various categories, such as glycan biosynthesis and metabolism, lipid metabolism, environmental information processing, genetic information processing, cellular processes, amino acid metabolism and other metabolism, for further discussion (Supplementary Table 4). The gene expression levels in these categories showed obvious differences under different temperature stresses, indicating the possible mechanisms of the strain Ant34-E75 response to different temperature stresses. Among the potential mechanisms, peptidoglycan biosynthesis, ABC transporters, lysine biosynthesis, phenylalanine metabolism, and nicotinamide and nicotinamide metabolism showed a marked increase at low temperature and a marked decrease at high temperature, while two-component systems, arginine and proline metabolism, cysteine and methionine metabolism and oxidative phosphorylation showed the opposite pattern. In addition, nitrogen metabolism increased markedly under both low- and high-temperature conditions. The results of the transcriptome sequencing and differential gene expression analyses provide valuable insights into the molecular mechanism of the Ant34-E75 response to temperature stress.

**Weighted gene co-expression network**. WGCNA was used to construct a gene co-expression network to identify the key molecular mechanism and identify novel functional genes involved in the response to temperature stress. The results indicated a scale-free topological fit index of 0.8, and a soft threshold of 18 was selected, resulting in an average connectivity of 53 (Fig. 3a). The analysis revealed positive correlations among the $OD_{600}$ (bacterial concentration), $OD_{490}$ (polysaccharide concentration) and temperature (Fig. 3b). In addition, among the seven modules, the blue module was markedly negatively correlated, while the brown module was markedly positively

**Table 2 Comparison of fatty acids of strains belonging to the same genus.**

| The fatty acids | 1 | 2 | 3 | 4 | 5 | 6 | 7 | 8 | 9 | 10 | 11 |
|---|---|---|---|---|---|---|---|---|---|---|---|
| iso-$C_{15:1}$G | 22.59 | 10.14 | 4.8 | 6.0 | 1.4 | 6.31 | 8.2 | | | | 2.0 |
| iso-$C_{15:0}$ | 21.66 | 25.96 | 40.3 | 43.2 | 32.8 | 27.89 | 22.4 | 68.8 | 7.6 | 16.3 | 34.2 |
| iso-$C_{17:0}$ 3-OH | 12.05 | 12.53 | 14.6 | | 19.0 | 15.96 | 16.4 | 0.8 | 2.0 | 2.3 | 29.4 |
| anteiso-$C_{15:0}$ | 7.26 | 14.43 | 7.3 | 21.3 | 2.4 | 5.08 | 3.3 | 8.4 | 15.7 | 20.7 | 2.9 |
| Summed Feature 3 | 4.17 | 1.79 | 3.2 | 2.4 | 9.9 | 6.85 | 8.8 | | | | 5.8 |
| iso-$C_{16:0}$ | 4.1 | 0.4 | 5.6 | 1.9 | 3.1 | 4.45 | 3.8 | 3.0 | 1.4 | 1.2 | 2.8 |
| $C_{17:0}$2-OH | 3.53 | | | 4.5 | | 0.88 | 2.3 | | | | 1.1 |
| Summed Feature 9 | 3.07 | 13.09 | 8.4 | | 16.8 | | 9.1 | | | | |
| iso-$C_{15:0}$ 3-OH | 2.95 | 2.24 | 2.2 | 2.1 | 3.5 | 2.60 | 3.2 | 1.6 | 5.4 | 1.7 | 4.3 |
| anteiso-$C_{17:1}$w9c | 2.53 | | | | | | | | | | |
| iso-$C_{16:0}$3-OH | 1.95 | 1.25 | 2.8 | 2.7 | 2.9 | 2.76 | 4.1 | | 9.2 | 2.1 | 2.3 |
| anteiso-$C_{15:1}$A | 1.81 | 1.14 | | | | | | | | | |
| $C_{16:0}$ | 1.78 | 1.96 | 2.0 | | 1.7 | 0.74 | 1.7 | 2.2 | 1.9 | 1.2 | 2.0 |
| $C_{15:0}$ 2-OH | 1.53 | 1.16 | | 1.1 | | 1.14 | 1.2 | | | | 0.6 |

The values are percentages of total fatty acids. Strains: 1, *Aequorivita marisscotiae* Ant34-E75; 2, *Aequorivita viscosa* 8-1b[T]; 3, *Aequorivita lutea* q18[T7]; 4, *Aequorivita sinensis* S1-10[T8]; 5, *Aequorivita iocasae* KX20305[T72]; 6, *Aequorivita aquimaris* D-24[T73]; 7, *Aequorivita echinoideorum* CC-CZW007[T74]; 8, *Aequorivita vladivostokensis* KMM 3516[T75]; 9, *Aequorivita antarctica* SW49[T4,9]; 10, *Aequorivita lipolytica* Y10-2[T4]; 11, *Aequorivita soesokkakensis* RSSK-12[T76].

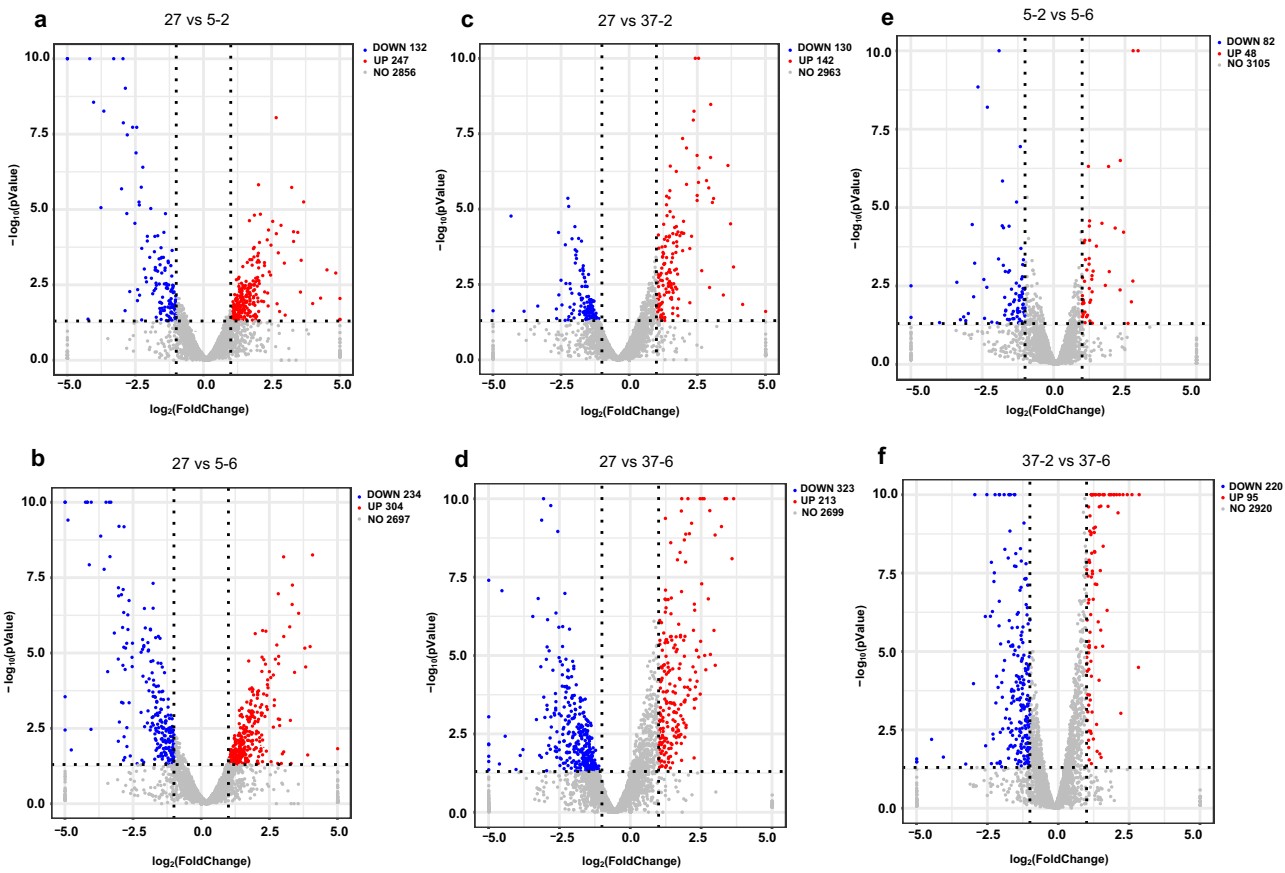

**Fig. 2 Volcano plot of the differentially expressed genes at different temperatures.** The X-axis indicates the fold change in gene expression (threshold, |log$_2$FoldChange| >1), while the Y-axis indicates the statistical significance level (threshold, $p$ value < 0.05). The strain was cultured to the logarithmic growth phase at 27 °C and named 27. Then, the bacteria were cultured at 5 °C for 2 h (5–2) and 6 h (5–6) and at 37 °C for 2 h (37–2) and 6 h (37–6).

correlated (Fig. 3c). To further analyse these findings, the top 20 DEGs with the highest weights in the blue and brown modules were selected and are listed in Supplementary Table 5. Most of the DEGs in the blue module were found to show decreased expression at low temperature and increased or unchanged expression at high temperature. In contrast, the brown module was characterized by DEGs that presented increased expression at low temperature and decreased or unchanged expression at high temperature. It could be concluded that more genes showed differential expression at low temperature than at high temperature. We noted that some novel hypothetical proteins were highly weighted in the brown module, such as E75_GM000872, E75_GM002528, E75_GM002833, E75_GM003171, and E75_GM000308. Other than these hypothetical proteins, *mur*C, which is an important gene in peptidoglycan biosynthesis, presented the highest weight.

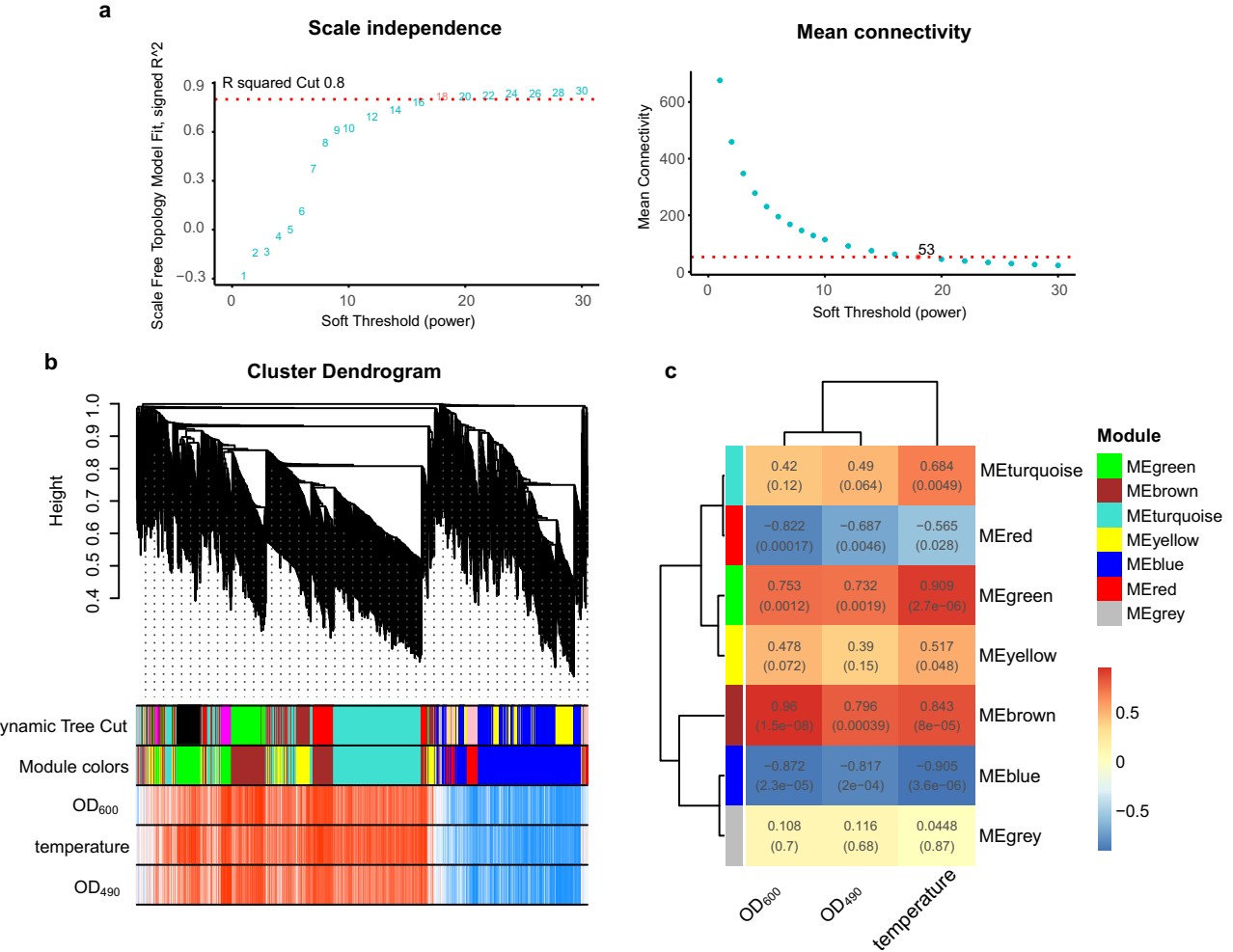

**Fig. 3 Gene coexpression network construction and module identification. a** Determination of soft-thresholding power in WGCNA. The numbers in the panel are the power candidates. **b** Hierarchical clustering dendrograms of identified DEGs. Modules correspond to branches and are labelled by colours as indicated by the first colour band underneath. After ME-based hierarchical clustering, the original modules are merged and presented in the second colour band. **c** Heatmap of module-trait associations. The correlation is provided in each cell, with fill colour as an indicator.

**Pathway prediction and qRT-PCR**. Further analysis of the DEGs related to peptidoglycan biosynthesis in the strain was conducted based on the annotation results and WGCNA. The results showed that the expression of DEGs involved in peptidoglycan synthesis increased at low temperature and decreased at high temperature (Supplementary Table 4). One of the genes with a high weight in the brown module, *mur*C, showed a marked positive correlation according to WGCNA (Supplementary Table 5). A peptidoglycan synthesis pathway was constructed based on these findings (Fig. 4).

To validate the results obtained from transcriptome sequencing, all the DEGs of the peptidoglycan synthesis pathway were subjected to qRT-PCR tests, which showed consistent results with the transcriptome data (Fig. 5). Although the transcriptome sequencing data indicated that the expression of DEGs for peptidoglycan synthesis increased at low temperature, it should be noted that transcriptome sequencing was performed at only 2 h and 6 h. The results showed that most changes occurred at 6 h rather than at 2 h. To further determine the role of peptidoglycan synthesis in the strain, qRT-PCR was performed after varying the treatment time. The results showed that the expression of those genes reached the peak of downregulated at 8 h under high temperature and upregulated at 2 h under low temperature (Fig. 5).

**Discussion**

In the culture experiment, the viscosity of the lipuid culture of the experimental bacterial strain was higher than that of other strains. This inference was made because the other strains were completely precipitated when centrifuged at 12,000 rpm for 10 min, while the experimental strain could not be completely precipitated, and orange liquid appeared between the bacteria and the water phase. We speculated that this might have been due to the production of exopolysaccharides by the strain. Exopolysaccharides have been shown to present various properties, such as antioxidant activity[22], immune modulation, antitumour activity[23], and cold resistance[24]. The experimental strain could still grow at 5 °C, which might indicate cold resistance, probably due to the production of extracellular polysaccharides. Our algal-bacterial symbiosis experiment also supported this hypothesis, as it showed an increased cold tolerance time at normal temperatures. Temperature is a very important environmental factor for all organisms. To adapt to temperature, cold-tolerant bacteria have developed special cell structures and enzyme metabolism mechanisms during their long-term biological evolution. It is well documented that the cold tolerance mechanisms of strains involve a complex regulatory process, which can be explained based on several aspects, such as membrane structure, cold-active enzymes, and protein components. However, there are still many

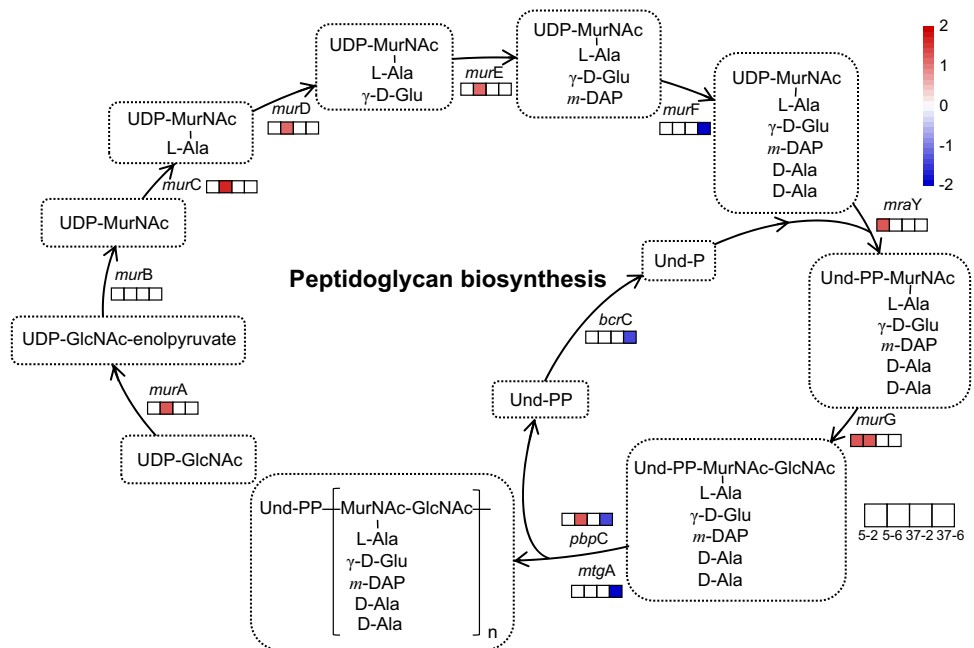

**Fig. 4 A proposed model summarizing the pathways for peptidoglycan biosynthesis involved in the response to temperature stress in the strain.** The four squares represent cells cultured at 5 °C for 2 h (5–2) and 6 h (5–6) and at 37 °C for 2 h (37–2) and 6 h (37–6). The log₂FoldChange represents the amount of gene expression compared to the optimal temperature (27 °C) under different temperature conditions with fill colour as an indicator.

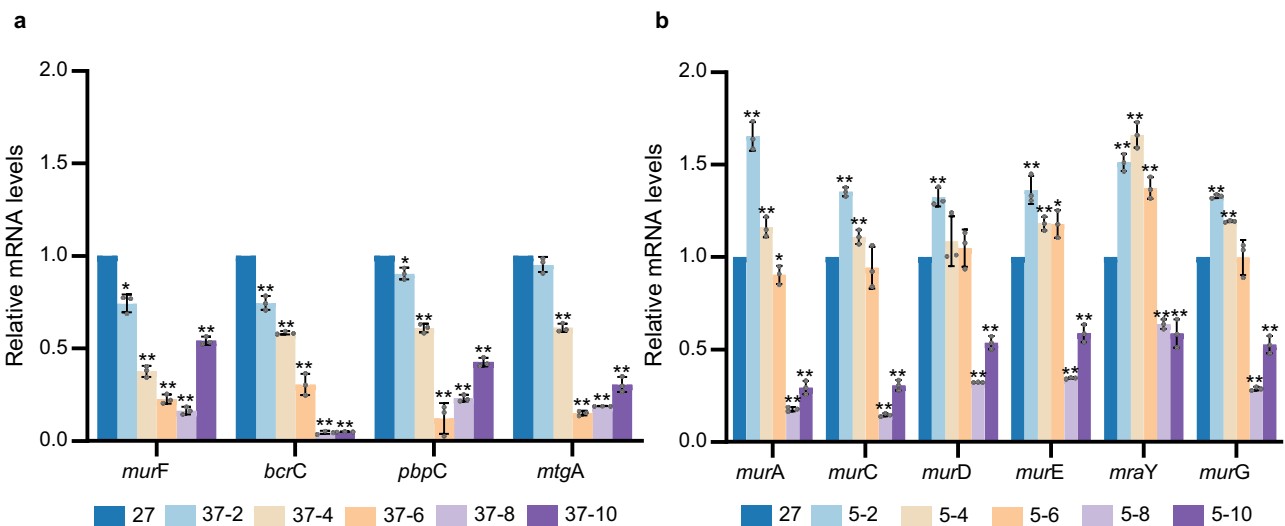

**Fig. 5 The expression levels of key genes of the peptidoglycan biosynthesis pathway were analysed by quantitative RT–PCR analysis; the Y-axis indicates the relative expression level; the X-axis indicates different genes.** The bar graphs were generated using data from three distinct biological replicates, visualized as circles. The error bars indicate the standard deviation. **a** Low-temperature treatment group. The strain was cultured to the logarithmic growth phase at 27 °C and then cultured at 5 °C for 2 h (5–2), 4 h (5–4), 6 h (5–6), 8 h (5–8), and 10 h (5–10). **b** The high-temperature treatment group. The strain was cultured to the logarithmic growth phase at 27°C and then cultured at 37 °C for 2 h (37–2), 4 h (37–4), 6 h (37–6), 8 h (37–8), and 10 h (37–10). Significant difference (*$p < 0.05$ and **$p < 0.01$).

microorganisms that have not been discovered in the deep-sea environment, and their temperature adaptation mechanisms are unknown and deserve further exploration.

Transcriptome sequencing revealed that the expression of differentially expressed genes involved in glycan biosynthesis and metabolism increased under low-temperature conditions but decreased under high-temperature conditions, with notable variation in the expression of peptidoglycan biosynthesis genes. The *mur*C gene of the peptidoglycan synthesis pathway showed high weight values in the brown module of WGCNA. Peptidoglycan is

a basic component of all bacteria and plays a role in maintaining bacterial morphology and adjusting osmotic pressure[25]. Suyal observed increased expression of peptidoglycan synthesis proteins in *Rhodococcus qingshengii* S10107 under low-temperature nitrogen deficiency[26]. As early as 1972, temperature-sensitive *Escherichia coli* strains were constructed by knocking out or replacing genes such as *mur*E[27], *mur*A and *mur*B[28]. A plasmid carrying the *mur*B gene can restore the temperature-sensitive growth of six *Staphylococcus aureus* mutants at a restrictive temperature[29]. Moreover, studies have shown that temperature

sensitivity can be weakened by restoring murA and/or murB in a temperature-sensitive mutant of *Corynebacterium glutamicum*[30]. These findings suggest that our experimental strain has adapted to temperature stress by increasing its peptidoglycan content under cold stress and decreasing it under heat stress, similar to how humans dress to survive in cold weather.

The differential gene expression of the ABC transport system, an important component responsible for the export of cell-surface glycoconjugates in both Gram-positive and Gram-negative bacteria[31], was found to be upregulated under low-temperature conditions but decreased under high-temperature conditions. In contrast, the expression of two-component system genes decreased at low temperature and increased at high temperature. Two-component systems play a role as thermosensors and regulate gene expression in microorganisms[32]. The phoR and phoP genes, which are involved in the response to phosphate limitation, were primarily upregulated in Ant34-E75 under these conditions[33], and they also presented a strong relationship with autophosphorylation at a conserved histidine residue[32]. Although PhoPR is essential for the normal growth and metabolism of a broad range of species, the regulatory functions and mechanisms of PhoP can vary in different bacteria[33]. Differential gene expression was also found to be clustered in a two-component system under different pH conditions in *Pseudoalteromonas agarivorans*[34]. Both the ABC transport system and the two-component system play important roles in the response to environmental changes, including temperature stress.

The ability of microorganisms to cope with temperature stress is largely dependent on their ability to maintain the activity of proteins required for growth. Studies have shown that the amino acid composition of thermophilic proteins exhibits certain general features, with many of the amino acids showing a marked temperature-dependent difference in frequency[35,36]. Our study showed a decrease in differential gene expression related to arginine and proline metabolism at low temperature and an increase at high temperature (Supplementary Table 5). Similarly, studies on other microorganisms, such as *Bacillus cereus*[37], *Shewanella gelidimarina*[38] and *Halomonas* sp.[39], have shown decreased arginine and proline contents at low temperature, which was also observed in the insect *Bactrocera dorsalis*[40]. This reduction in arginine and proline content is thought to enhance protein conformational flexibility and improve enzyme activity at low temperatures by reducing hydrogen bonds and salt bridges. In contrast, the contents of lysine and glutamine increase under such conditions[41], as observed in our experiment. The results also showed a consistent trend of increased expression of serine and threonine in addition to the downregulation of arginine and proline[38]. At high temperature, the biosynthesis of tyrosine and tryptophan was markedly upregulated, while arginine biosynthesis was downregulated, which may be related to the adjustment of G + C contents, with a higher G + C content being beneficial for the maintenance of protein structure.

Temperature stress generally leads to limited microbial growth and inhibited metabolism. However, our results showed that the nitrogen metabolism of Ant34-E75 was markedly upregulated at both low temperatures and high temperatures, which could be due to utilization of additional nitrogen sources. Besides, oxidative phosphorylation was markedly up-regulated at high temperature, while nicotinate and nicotinamide metabolism were markedly up-regulated at low temperature, which may play a role in the adaptation of microorganisms to temperature. And last but not least, the WGCNA screened several genes with unknown functions, newly perspective genes, which are most likely related to the temperature adaptation of warrant further investigation.

## Methods

**Strain source and culture medium**. The strain investigated in this study, designated Ant34-E75, was isolated from deep-sea sediment collected during the 34th Chinese National Antarctica Expedition. The samples were collected from waters of the Antarctic and the southern end of the Atlantic Ocean in the Scotia Sea at 37°32.471' W, 60°23.008' S and a water depth of 2560 m. *Aequorivita viscosa* 8-1b$^T$ (= CGMCC 1.11023), which was obtained from the China General Microbiological Culture Collection Centre (CGMCC), was employed as a reference strain in this study. The cultivable strain was obtained through a combination of gradient dilution coating and streaking inoculation techniques on marine ZoBell 2216E medium (peptone 5 g; yeast extract 1 g; filtered seawater: ultra-pure water (v/v) = 2: 1).

**Characterization of strain Ant34-E75**. The identification of strain Ant34-E75 was conducted by evaluating its growth on marine ZoBell 2216E medium at different temperatures (5 °C, 15 °C, 27 °C, and 37 °C), pH levels (pH 4.0–13.0 with intervals of 1 units)[42], and NaCl concentrations (0–10% in 2% increments, w/v). In order to observe the cells, the culture solution of the strain Ant34-E75 was fixed with 1% glutaraldehyde buffer, rinsed with ethanol and then freeze-dried. The S4800 cold field scanning electron microscope (SEM, Hitachi S-4800, Tokyo, Japan) was used for observation. The degradation of gelatine, casein, sodium alginate, urea, starch, Tween 80 and xylan was tested on ZoBell 2216E plates supplemented with appropriate substrates as described previously of the strain Ant34-E75[43]. Catalase activity was determined by bubble production in 3% (v/v) $H_2O_2$, and oxidase activity was determined using 1% (w/v) N'N'N'N'-tetramethyl-p-phenylenediamine of the strain Ant34-E75. Other biochemical tests and enzyme assays of the strain Ant34-E75 were performed using the API 20NE and API ZYM kits (bioMérieux) and GN2 MicroPlates (Biolog) according to the manufacturers' instructions. Respiratory quinones of the strain Ant34-E75 were analysed by using High Performance Liquid Chromatography[44]. The fatty acids from freeze-dried biomass of the strain Ant34-E75 and *Aequorivita viscosa* 8-1b$^T$ were extracted, methylated and then analyzed using an Agilent model 6890 A gas chromatograph and the Sherlock Microbial Identification System (MIDI, Inc.) version 6.3 with the database RTSBA6[45,46]. Polar lipids of the strain Ant34-E75 were determined using two-dimensional thin layer chromatography (TLC) on silica gel thin layers[47]. The TLC plates were developed in chloroform/methanol/water (65 : 25 : 4, by vol.) in the first direction, followed by chloroform/methanol/acetic acid/water (80 : 12 : 15 : 4, by vol.) in the second direction[48].

The 16 S rRNA sequence of strain Ant34-E75 was aligned via a BLAST search in NCBI (https://blast.ncbi.nlm.nih.gov/Blast.cgi). Phylogenetic trees were reconstructed using the neighbour-joining[49] and maximum-likelihood[50] algorithms based on the 16 S rRNA sequences using *Escherichia coli* as an outgroup genus with MEGA X[51]. the best DNA model was selected by MODELS in MEGA X was the Kimura 2-parameter model (K2)[52] and the Rates among Sites was Gamma Distributed With Invariant Sites (G + I). The maximum likelihood tree of 16 S rRNA was constructed with a bootstrap value of 1000[8] via model K2 + G + I.

**Whole-genome sequencing and assembly**. The strain Ant34-E75 was cultivated at 27°C and 150 r/min to exponential phase and harvested by centrifugation at 4°C (12000 × g, 20 min). The SDS method was used to extract the genomics DNA of the strain Ant34-E75 according to Xia et al.[53]. Then 10 K SMRT Bell library was built by the SMRT Bell TM Template kit (version 1.0) and a

350 bp small-fragment library was built by the NEBNext®Ultra™ DNA Library Prep Kit for Illumina (NEB, USA), which quantification were all performed by a Qubit. The two libraries were sequenced using PacBio RS II SMRT and Illumina NovaSeq PE150 at Beijing Novogene Bioinformatics Technology Co., Ltd. (P.R. China). Raw reads were filtered to obtain clean reads, which were assembled using SMRT Link v5.0.1 software and optimized assembly results using Arrow software (version 2.2.1)[54,55]. Encoding gene predictions were performed using GeneMarkS software (version 4.17)[56]. The tRNAscan-SE (version 1.3.1)[57] and rRNAmmer software (version 1.2)[58] were used to perform the sequencing of tRNA and rRNA. ANI analysis was conducted on ANI Calculator[59] (https://www.ezbiocloud.net/tools/ani), and the dDDH analysis was calculated by the Genome-to-Genome Distance Calculator[60] (https://ggdc.dsmz.de/ggdc.php#).

**Algae-bacteria symbiosis**. The culture solution of the strain Ant34-E75 (1 mL) was centrifuged, and the supernatant was discarded. The bacteria were then resuspended in 1 mL of Tris-Acetate-Phosphate (TAP) liquid medium[61]. *Chlamydomonas reinhardtii* (2 mL) grown to the logarithmic phase was centrifuged, and the algae liquid was resuspended in 1 mL of TAP liquid medium. The algae and bacteria were mixed at proportions of 1%, 10% and 20% (ratio by volume) and then cultured in a light incubator at 26°C (optimum growth temperature of *Chlamydomonas reinhardtii*) for 1.5 h. The coculture mixture was treated at -20°C for different times (10 min, 20 min, 30 min, 40 min, and 50 min). Finally, 6 µL of the treated mixture was inoculated into TAP solid medium and incubated at 26°C for 5 days at a light intensity of $50 \pm 10$ µEm$^{-2}$s$^{-1}$.

**Sequencing for transcriptomic analysis**. The strain was grown to the logarithmic growth phase at 27°C and then treated at 5 °C or 37 °C for 2 h and 6 h (Fig. 1d). Triplicate samples were prepared for each condition. The Trizol reagent was used to obtain total RNA of strain Ant34-E75, and Qubit RNA Assay Kit in Qubit 2.0 fluorometer (Shenzhen, Shanghai) was used to detect RNA concentration. RNA integrity and genomics contamination were detected on agarose gel. To obtain the DEGs of the strain under different temperature conditions, we sent the samples to Sangon Biotech (Shanghai) Co., Ltd. for transcriptome sequencing using the Illumina HiSeq. The raw data were evaluated for quality using FastQC (version 0.11.2) and cut by Trimmomatic (version 0.36)[62]. To validate the accuracy of the transcriptome data, the processed data were compared to the reference genome (the complete genome of the strain Ant34-E75) by Bowtie2 (version 2.3.2)[63]. Then, the processed data were assembled according to the reference genome using Rockhopper (version 2.0.3)[64]. Finally, differential gene expression analysis was performed using DESeq2 (version 1.12.4)[65] and the DEGs were identified with a threshold of adjusted *p* value < 0.05. For functional annotation of differential genes, topGO (version 2.24.0) was used for Gene Ontology (GO) annotation, and clusterProfiler (version 3.0.5) was used for Kyoto Encyclopedia of Genes and Genomes (KEGG)[66] pathway and Clusters of Orthologous Groups (COG)[67] classification annotation.

**Weighted gene co-expression network construction**. WGCNA analysis was used to construct a coexpression network for the filtered genes[68]. The gene expression matrices of all the DEGs and sample traits, including OD$_{600}$ (bacterial concentration), OD$_{490}$ (polysaccharide content) and temperature, were used for WGCNA. The polysaccharide content was determined by the acetone sulphuric acid method[69]. After sample clustering, scale independence and mean connectivity analysis of modules with different power values was performed to determine the soft threshold for module analysis. The power value was set from 1 to 30, and the soft threshold was determined when the scale independence value reached 0.8[70]. The relationship between module and trait was calculated, and genes in marked modules were exported for further analysis.

**Quantitative real-time RT−PCR analysis**. The transcriptome data were validated using real-time quantitative PCR. The treatment conditions were the same as those for transcriptome sequencing: the strain was grown to the logarithmic growth phase at 27 °C and then treated at 5 °C or 37 °C. However, a greater number of treatment times at different temperatures were examined, including 2 h, 4 h, 6 h, 8 h, and 10 h. The experiment was performed in triplicate. The 16 S RNA gene was used as the reference gene for normalization[71]. The primers used (Supplementary Table 6) in the experiments were designed according to the functional gene sequence using Primer6.0. Total RNA was isolated using the *TransZol Up Plus RNA Kit* (ER501-01-V2). The samples under different conditions were lysed with *TransZol Up*. After RNA Extraction Agent was added, the solution separates into an upper colourless aqueous phase (containing RNA), an interphase and a lower pink organic phase. A silica-based spin column was used to specifically bind to RNA in the aqueous phase. And transcribed to cDNA using a One-Step gDNA Removal kit (AT341-02). The 2$^{-\triangle\triangle Ct}$ procedure was applied to evaluate the qRT−PCR data (Supplementary Table 7).

**Statistics and reproducibility**. In the bacterial physiology experiments, every group was composed of three replicates. Transcriptome sequencing was conducted for three samples within each group, and the raw data for each of these samples has been deposited into the Genome Sequence Archive (GSA). Similarly, qRT-PCR was performed with three replicates, and data analysis using SPSS included the calculation of significance levels through independent sample *t* tests (*$p < 0.05$ and **$p < 0.01$).

**Reporting summary**. Further information on research design is available in the Nature Portfolio Reporting Summary linked to this article.

## Data availability

The original contributions presented in the study are publicly available. The whole genome sequence data were available in the National Centre for Biotechnology Information (NCBI) database under accession number CP122379 and the GenBank accession number for the 16 S rRNA gene sequence was OR529246. The source data for the transcriptomic results were also deposited in Genome Sequence Archive (GSA) under the accession number: CRA011001. The strain Ant34-E75 ( = CCTCC AB 2023079 = KCTC 92993) was deposited into the general collection of microorganism of the China Centre for Type Culture Collection (CCTCC) and the Korean Collection for Type Cultures (KCTC). The source data behind Figs. 2, 3 and 4 can be found in Supplementary Data 1, and the source data behind Fig. 5 can be found in Supplementary Table 7.

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

## Acknowledgements

This work was financially supported by the Chinese National Natural Science Foundation (Grant No: 41006102) and The China-ASEAN Blue Partnership Construction Program, grant/award numbers: WJ1324013 and WJ1324015.

## Author contributions

Authors W.L. and B.C. conceived and designed the study, contributed equally to this work. J.L. helped to measure the biochemical characteristics of the strains. S.L. helped to perform the transcriptome analysis. A.D. provided materials and opinions for the experiment of algae-bacteria symbiosis. L.Z. coordinated the whole genome sequencing. W.L. wrote most of the manuscript with major review and editing by B.C. and J.L. All authors read and approved the final manuscript.

## Competing interests

The authors declare no competing interests.
