## [Peer Review File · Communications Biology]

Reviewers' comments:

Reviewer #1 (Remarks to the Author):

The study by Liu et al investigated the "Taxonomic identification and temperature stress tolerance mechanisms of *Aequorivita scotiaensis* sp. nov."

The authors described the new member of the genus *Aequorivita* and the mechanism of molecular adaptation using genomics and transcriptomics approach. However, the manuscript can be improved extensively, especially in the introduction and discussion sections will help the readers for a better understanding of the manuscript. I also suggest the authors to cite recent papers on mechanisms of cold adaptation in bacteria and their symbiotic association. Therefore, I found the study is interesting however, needs major revisions. Below are my comments

Line 13: ...16S rDNA gene sequence similarity of 97.18% with the strain *Aequorivita viscosa* 8-1bT

Line 15-16: Either delete the sentence or provide reason briefly why *Chlamydomonas reininatus* used here

Line 26-49: The information provided here are indistinguishable or very general. In many places I could see no connectivity between sentences which lacks readability. To make it perfect, delete most commonly used sentences (e.g. The Ocean covers almost 75% of the Earth's surface...) and provide only necessary information which is relevant to the paper.

Line 71: What is MRSA? Provide full name for the abbreviation at the first mention

In my opinion, the introduction part is irrational and has to be re-written because in most of the places I could see only results, discussion and summary parts rather than actual introduction part. The most important component "Aim and Objectives of the study" is missing.

Line 93 "investigated in this strain" typo error

Line 117: Please include accession number of 16S rRNA and length of nucleotides obtained (it should be more than 1400bp)

Line 120: Along with ANI, authors should also include DNA-DNA hybridization reports with the closest type strains when 16S rRNA gene similarity is above 97%. In this report, *Aequorivita scotiaensis* is shown a similarity of 97.18% to the reference strain *Aequorivita viscosa* 8-1b. Since the authors already sequenced whole genome, they can perform digital DNA-DNA hybridization with the closest strain *Aequorivita viscosa* 8-1b and include in the results section.

Line 128: It is mandatory to add nomenclature information under subheading Description of *Aequorivita scotiaensis* sp. nov.

Line 141-143 Authors could describe in detail about the genome features includes of protein-coding genes,... etc..

Line 165 what was the reference genome used in the transcriptome analysis?

Line 329: Subheading 'Strain identification' not suitable for this part. The data shows physiology, biochemical characterization, molecular identification and chemotaxonomy as well. Hence authors needs to find alternate subheading, or split the parts as mentioned above and include more information on the methods adopted. For example, in Line 340, how fatty acid methyl esters are identified, what method and which library used? In addition, it should be clear whether all the experiments are conducted parallel with closest type strain *Aequorivita viscosa* 8-1b.

Line 330-332: It is not clear whether growth experiments conducted in which medium (broth or agar plates) and the buffers used to adjust wide range of pH

Line 343: Include references for MEGA X, phylogenetic trees, alignments, bootstrap etc.

Line 361 what is the composition of TAP liquid medium

How the strain maintained after isolation? Authors mentioned samples collected from 2560 m depth Which platform was used for the whole genome sequencing. Authors could describe how the library preparation and sequencing was done?

Line 360 What do you mean by bacterial solution?

Line 371 Sequencing for transcriptomic analysis: This section is not clear to me how the RNA extraction done? Add the references for the softwares/tools used. Mention the settings in the tool used during the analysis of the transcriptome.

Line 441: Add methodology for SEM analysis in methods section.

Table 1: What is 5-37, 27? Is the growth temperature range and optimum temperature? If so, please provide the details for better readability (also for other parameters like, NaCl, pH)

Table 1 & Table 2: Authors should clearly mention which data generated during this study and which data obtained from references.

Figure 1a: Which method used for constructing phylogenetic tree?

Reviewer #2 (Remarks to the Author):

The manuscript is quite interesting and the work done comprehensive. Conclusions are well-supported by the data.

I have made corrections on a Word version of the manuscript. Further comments are below. The main comments are around the taxonomic description. Other comments are minor in nature

Line 21 "Furthermore, it enriches our understanding of the species and genetic diversity of deep-sea microorganisms." Suggest deleting these words since the previous sentence sufficiently concludes the summary of the study.

Line 63 and elsewhere. Use the term Bacteroidota, Flavobacterium-Cytophaga is rather out-dated. e.g. Members of the phylum Bacteroidota...

For taxonomic names the rank should be mentioned first then the name e.g. the genus *Aequorivita*.

Line 81 What is meant by "traditional MRSA activity-guided fractionation"? Assuming MRSA stands for methicillin resistant *Staphylococcus aureus* I assume fractionation is guided by antimicrobial activity? I think the sentence could be rewritten for. Improved clarity

Line 121 , Line 151-166. If the goal is to describe a new species you should mention that the strain studied becomes the new species' type strain. Also the description must explicitly mention the strain studied as the type strain otherwise the new species will not be recognized as a valid description by the ICSP (International Committee of Systematics of Prokaryotes). The species description needs to include the etymology of the species name *scotaiensis*. See LPSN: www.bacterio.net for more information including many examples of etymology of species names. Also the strain needs to be deposited into at least two culture collections recognized by the WFCC (World Federation of Culture Collections) to be considered valid as well as being mentioned in a Validation List in the journal IJSEM (for this part the new name will be subject to nomenclatural review and you need to email the Validation list editor). Good cultures collections in China include CCTCC and CGMCC but a culture collection in another country should also host the strain e.g. KCTC in Korea for example. If the strain needs to be restricted (due to patents) there should be guidelines available for this (e.g. WFCC and the WIPO Budapest Treaty). An alternative is to describe the strain under the SeqCode (see Hedlund et al. 2022 <https://www.nature.com/articles/s41564-022-01214-9>).

Line 156 Do you mean 4-nitrophenyl-beta-D-galactopyranoside?

Line 222 Unclear on the relevance of optical density measurements in the co-expression network analysis. More explanation needed or perhaps just mention temperature.

Line 254 Instead of "resumed" do you mean "plateaued", i.e. levelled off?

Line 402-404 A brief summary of the RNA extraction process used would be a useful addition to the methods section.

Response to Reviewers

At the very beginning, I would like to show my gratitude for the reviewers and editor to give us the pertinent and useful comments. I have revised the manuscript according to most of the comments.

-Reviewer 1

1. Line 13: ...16S rDNA gene sequence similarity of 97.18% with the strain *Aequorivita viscosa* 8-1bT.

Response: It has been revised in line 14.

2. Line 15-16: Either delete the sentence or provide reason briefly why *Chlamydomonas reinhardtii* used here

Response: *Chlamydomonas reinhardtii* is often used as a model organism in various studies due to its short growth cycle, moderate cell size and clear genetic background. It has been supplemented in line 17.

3. Line 26-49: The information provided here are indistinguishable or very general. In many places I could see no connectivity between sentences which lacks readability. To make it perfect, delete most commonly used sentences (e.g. The Ocean covers almost 75% of the Earth's surface...) and provide only necessary information which is relevant to the paper.

Response: It has been deleted in lines 27-49.

4. Line 71: What is MRSA? Provide full name for the abbreviation at the first mention

Response: The MRSA has been supplemented in lines 72-73.

5. In my opinion, the introduction part is irrational and has to be re-written because in most of the places I could see only results, discussion and summary parts rather than actual introduction part. The most important component "Aim and Objectives of the study" is missing.

Response: The introduction has been substantially revised in response to the recommendations in lines 27-120.

6. Line 93 "investigated in this strain" typo error

Response: It has been revised in line 105.

7. Line 117: Please include accession number of 16S rRNA and length of nucleotides obtained (it should be more than 1400bp)

Response: It has been revised in lines 131-134. The strain Ant34-E75's accession number of 16S rRNA is OR529246, and length of nucleotides obtained are 1,430 bp. The strain *Aequorivita viscosa* 8-1b^T was downloaded from NCBI.

8. Line 120: Along with ANI, authors should also include DNA-DNA hybridization reports with the closest type strains when 16S rRNA gene similarity is above 97%. In this report, *Aequorivita scotiaensis* is shown a similarity of 97.18% to the reference strain *Aequorivita viscosa* 8-1b. Since

the authors already sequenced whole genome, they can perform digital DNA-DNA hybridization with the closest strain *Aequorivita viscosa* 8-1b and include in the results section.

Response: It has been supplemented in lines 137-140. The strain Ant34-E75 performed digital DNA-DNA hybridization (dDDH) analysis with *Aequorivita viscosa* 8-1b^T. The results showed that the dDDH was 20.0%, which was lesser than 70%.

9. Line 128: It is mandatory to add nomenclature information under subheading Description of *Aequorivita scotiaensis* sp. nov.

Response: It has been revised in line 146. We sent an email to International Journal of Systematic and Evolutionary Microbiology (IJSEM) about the naming of the strain and supplemented the information in the manuscript.

10. Line 141-143 Authors could describe in detail about the genome features includes of protein-coding genes,... etc..

Response: It has been supplemented in lines 137-140. The genome size of the type strain is 3,451,112 bp with a G+C content of 38.41mol%, and the predicted number of coding genes was 3235, including 37 tRNA, 2 5S rRNA, 2 16S rRNA, 2 23S rRNA and 1 sRNA. The complete genome sequence has been deposited in the NCBI database under accession number CP122379, and the GenBank accession number for the 16S rRNA gene sequence was OR529246.

11. Line 165 what was the reference genome used in the transcriptome analysis?

Response: It has been supplemented in lines 191-192. The reference genome is the complete genome of the strain Ant34-E75.

12. Line 329: Subheading 'Strain identification' not suitable for this part. The data shows physiology, biochemical characterization, molecular identification and chemotaxonomy as well. Hence authors needs to find alternate subheading, or split the parts as mentioned above and include more information on the methods adopted. For example, in Line 340, how fatty acid methyl esters are identified, what method and which library used?

Response: Subheading has been modified to 'Characterization of strain Ant34-E75' in line 358, if you think this topic is appropriate. More information on the methods adopted has been supplemented in lines 359-383. The fatty acids from freeze-dried biomass were extracted, methylated and then analyzed using an Agilent model 6890A gas chromatograph and the Sherlock Microbial Identification System (MIDI, Inc.) version 6.3 with the database RTSBA6.

13. In addition, it should be clear whether all the experiments are conducted parallel with closest type strain *Aequorivita viscosa* 8-1b.

Response: It has been supplemented in lines 359-383, and the reference dates were also mentioned in the Legends of Tables 1 and 2. During the review process, we found differences in the fatty acid profiles of the strains reported by different researchers, while the other aspects of physiology and biochemistry did not show significant differences. Therefore, we only tested the fatty acids of both Ant34-E75 and *Aequorivita viscosa* 8-1b^T.

14. Line 330-332: It is not clear whether growth experiments conducted in which medium (broth or

agar plates) and the buffers used to adjust wide range of pH

Response: It has been supplemented in line 360. The growth experiments were conducted in marine ZoBell 2216E medium. The method of adjusting pH was referred to the reference, which has been cited in the manuscript. The buffers we used were as follows: pH 4.0 – 5.0: 0.1 M citric acid/0.1 M sodium citrate; pH 6.0 – 8.0: 0.1 M KH₂PO₄/0.1 M NaOH; pH 9.0 – 10.0: 0.1 M NaHCO₃/0.1 M Na₂CO₃; pH 11.0: 0.05 M Na₂HPO₄/0.1 M NaOH; pH 12.0 – 13.0: 0.2 M KCl/0.2 M NaOH.

15. Line 343: Include references for MEGA X, phylogenetic trees, alignments, bootstrap etc.

Response: It has been supplemented in lines 384-391.

16. Line 361 what is the composition of TAP liquid medium

Response: It has been supplemented in line 419. TAP liquid medium, which stands for Tris-Acetate-Phosphate liquid medium, is a common medium used for *Chlamydomonas reinhardtii*. This medium was derived from the work of Gorman, D.S., and R.P. Levine (1965), and we have included a quotation of their study. The TAP liquid medium used in this experiment was prepared by purchasing TAP salts, Phosphate solution, Hutner's trace elements, Glacial acetic acid, and Tris base.

17. How the strain maintained after isolation? Authors mentioned samples collected from 2560 m depth

Response: The deep-sea sediment, from which strain Ant34-E75 was isolated, was collected during the 34th Chinese National Antarctica Expedition for the purpose of diversity analysis of culturable microorganisms. During the process of culturing microorganisms, we directly used the marine 2216E medium at 15 °C, without simulating the deep-sea environment conditions such as pressure.

18. Which platform was used for the whole genome sequencing. Authors could describe how the library preparation and sequencing was done?

Response: It has been supplemented in lines 394-409.

19. Line 360 What do you mean by bacterial solution?

Response: It has been supplemented in line 417. The bacterial solution in line 417 is culture solution of the strain Ant34-E75.

20. Line 371 Sequencing for transcriptomic analysis: This section is not clear to me how the RNA extraction done? Add the references for the softwares/tools used. Mention the settings in the tool used during the analysis of the transcriptome.

Response: It has been supplemented in lines 430-447.

21. Line 441: Add methodology for SEM analysis in methods section.

Response: It has been supplemented in lines 362-365.

22. Table 1: What is 5-37, 27? Is the growth temperature range and optimum temperature? If so, please provide the details for better readability (also for other parameters like, NaCl, pH)

Response: It has been supplemented in Table 1. The 5-37, 27 is the growth temperature range and optimum temperature.

23. Table 1 & Table 2: Authors should clearly mention which data generated during this study and which data obtained from references.

Response: The reference dates were also referenced in the Legend of the Table 1 (522-524) and 2 (530-532).

24. Figure 1a: Which method used for constructing phylogenetic tree?

Response: It has been supplemented in line 480. The Figure 1a shows the maximum-likelihood phylogenetic tree.

-Reviewer 2

1. I have made corrections on a Word version of the manuscript. Further comments are below. The main comments are around the taxonomic description. Other comments are minor in nature

Response: The manuscript has been modified according to your Word version of the manuscript.

2. Line 21 “Furthermore, it enriches our understanding of the species and genetic diversity of deep-sea microorganisms.” Suggest deleting these words since the previous sentence sufficiently concludes the summary of the study.

Response: It has been deleted in lines 23-24.

3. Line 63 and elsewhere. Use the term Bacteroidota, Flavobacterium-Cytophaga is rather out-dated. e.g. Members of the phylum Bacteroidota...

For taxonomic names the rank should be mentioned first then the name e.g. the genus *Aequorivita*.

Response: It has been deleted. The introduction has been revised to be more concise according to the comments of reviewers.

4. Line 81 What is meant by “traditional MRSA activity-guided fractionation”? Assuming MRSA stands for methicillin resistant *Staphylococcus aureus* I assume fractionation is guided by antimicrobial activity? I think the sentence could be rewritten for. Improved clarity

Response: The MRSA has been supplemented in lines 72-73. And the sentence has been rewritten.

5. Line 121, Line 151-166. If the goal is to describe a new species you should mention that the strain studied becomes the new species' type strain. Also the description must explicitly mention the strain studied as the type strain otherwise the new species will not be recognized as a valid description by the ICSP (International Committee of Systematics of Prokaryotes). The species description needs to include the etymology of the species name *scotaiensis*. See LPSN: www.bacterio.net for more information including many examples of etymology of species names. Also the strain needs to be deposited into at least two culture collections recognized by the WFCC (World Federation of Culture Collections) to be considered valid as well as being mentioned in a Validation List in the journal IJSEM (for this part the new name will be subject to nomenclatural review and you need to email the Validation list editor). Good cultures collections in China include CCTCC and CGMCC but a culture collection in another country should also host the strain e.g. KCTC in Korea for example. If the strain needs to be restricted (due to patents) there should be guidelines available for this (e.g. WFCC and the WIPO Budapest Treaty). An alternative is to describe the strain under the SeqCode

(see Hedlund et al. 2022 <https://www.nature.com/articles/s41564-022-01214-9>).

Response: Thank you for your comments, and we have supplemented the type strain information in line 111, 142, and 163-165. We also sent an email to the International Journal of Systematic and Evolutionary Microbiology (IJSEM) regarding the naming of the strain and provided additional information in line 146-147. The strain Ant34-E75 has been deposited into the China Center for Type Culture Collection (CCTCC) and the Korean Collection for Type Cultures (KCTC), as stated in the Data availability section, and we have included this information in line 163.

6. Line 156 Do you mean 4-nitrophenyl-beta-D-galactopyranoside?

Response: It has been revised in line 153.

7. Line 222 Unclear on the relevance of optical density measurements in the co-expression network analysis. More explanation needed or perhaps just mention temperature.

Response: It has been revised in line 218-219. The OD₆₀₀ represents the concentration of the culture solution, which is used to assess the growth of strains at different temperatures. The OD₄₉₀ represents the concentration of total polysaccharides. Our speculation was that the increased cold tolerance of *Chlamydomonas reininatus* strain may be due to an increase in polysaccharide concentration. We added the OD₄₉₀ measurement to test this hypothesis. However, the results did not show a significant increase in polysaccharide concentration under low temperature conditions in this experiment. This could potentially be attributed to the large difference in culture solution concentration between different temperatures. Consequently, both OD₆₀₀ and OD₄₉₀ measurements were included for further analysis in the WGCNA. The analysis revealed positive correlations among bacterial concentration (OD₆₀₀), polysaccharide concentration (OD₄₉₀), temperature, and the brown module.

8. Line 254 Instead of “resumed” do you mean “plateaued”, i.e. levelled off?

Response: It has been supplemented in line 251-253. The previous sentence was not precise enough.

9. Line 402-404 A brief summary of the RNA extraction process used would be a useful addition to the methods section.

Response: It has been supplemented in line 430-434.

Reviewers' comments:

Reviewer #1 (Remarks to the Author):

Dear Editor and authors

Authors have made necessary corrections in the revised manuscript. The revised manuscript can be accepted.

Reviewer #2 (Remarks to the Author):

Some further minor changes should be completed:

Line 32 and Line 35 Spelling: Aequorivita.

Line 37 Replace "modified" with "emended".

Line 52 Italicise the name *Staphylococcus aureus*.

Line 73more extracellular carbohydrates and polymeric substances...

Line 99 Replace "under" with "within".

Line 108 replace sentence here: "Digital DNA-DNA hybridization (dDDH) was performed to determine the similarity of the genomes of strain Ant34-E75 and *A. viscosa* 8-1bT.

Line 122 Positive for liposterase C8...

Line 134 Change 38.41 mol% to 38.4 mol%

Line 143 Change sentence here ...could grow after 10 min of cold treatment but could not grow if the treatment was increased to 20 min, when strain Ant34-E75 was inoculated at either 0% or 1%.

Response to Reviewers

At the very beginning, I would like to show my gratitude for the reviewers and editor to give us the pertinent and useful comments. I have revised the manuscript according to most of the comments.

-Reviewer 1

Thank you very much for taking the time to review our manuscript. Your insights and suggestions have been extremely valuable, and they have significantly contributed to improving the quality of the paper. We have learned a great deal from your feedback.

-Reviewer 2

Thank you very much for your suggestions. These are issues we might not have noticed on our own, and your advice has greatly improved the quality of the paper.

1. Line 32 and Line 35 Spelling: Aequorivita.

Response: It has been revised in line 32 and 35.

2. Line 37 Replace “modified” with “emended”.

Response: It has been revised in line 37.

3. Line 52 Italicise the name *Staphylococcus aureus*.

Response: It has been revised in line 52.

4. Line 73 ...more extracellular carbohydrates and polymeric substances...

Response: It has been revised in line 73.

5. Line 99 Replace “under” with “within”.

Response: It has been revised in line 99.

6. Line 108 replace sentence here: “Digital DNA-DNA hybridization (dDDH) was performed to determine the similarity of the genomes of strain Ant34-E75 and *A. viscosa* 8-1bT.

Response: It has been revised in line 108-110.

7. Line 122 Positive for lipoesterase C8....

Response: It has been revised in line 125.

8. Line 134 Change 38.41 mol% to 38.4 mol%

Response: It has been revised in line 137 and Table 1.

9. Line 143 Change sentence here ...could grow after 10 min of cold treatment but could not grow if the treatment was increased to 20 min, when strain Ant34-E75 was inoculated at either 0% or 1%.

Response: It has been revised in line 146-148.

-Reviewer 3

We sincerely apologize for our oversight in the previous revision. Regrettably, we immediately incorporated the feedback from the last two reviewers without adequately addressing your valuable comments. We deeply apologize for this oversight.

1. I suggest a phylo/taxonomic approach, best using the GTDB-tk pipeline, also to get potential information on other genomes of the genus.

Response: It has been supplemented in line 111-113. We are aware that GTDBtk is a common software for metagenomic analysis, but it also serves as a method for microbial taxonomy. Following your suggestion, we employed GTDBtk to classify strain A, and the results indicate that strain A belongs to the following taxonomic ranks: d__Bacteria; p__Bacteroidota; c__Bacteroidia; o__Flavobacteriales; f__Flavobacteriaceae; g__Aequorivita; s__.

2. It appears that genomes from 11 different taxonomically validly published *Aequorivita* species are available, but you included only six of them in your average nucleotide identity analysis. Therefore it is necessary to include the genomes of the other five species as well (*A. vladivostokensis*, *A. aquimaris*, *A. sinensis*, *A. iocasae*, *A. echinoideorum*).

Response: It has been supplemented in Supplementary Table 1. At the outset, we initially selected a few strains based on their 16S rRNA sequences similarity for ANI analysis. In light of your suggestion, we have made additional inclusions, and the list now comprises the following strains: 1, *Aequorivita marisscotiae* Ant34-E75; 2, *Aequorivita viscosa* 8-1b^T; 3, *Aequorivita lutea* q18^T; 4, *Aequorivita sinensis* S1-10^T; 5, *Aequorivita iocasae* KX20305^T; 6, *Aequorivita aquimaris* D-24^T; 7, *Aequorivita echinoideorum* CC-CZW007^T; 8, *Aequorivita vladivostokensis* KMM 3516^T; 9, *Aequorivita antarctica* SW49^T; 10, *Aequorivita lipolytica* Y10-2^T; 11, *Aequorivita soesokkakensis* RSSK-12^T.

3. In addition, try to add as many physiological data of all *Aequorivita* species in the physiological table.

Response: It has been supplemented in Table 1 and Table 2. Based on the strains selected for the supplemented ANI analysis, we have also added their fundamental characteristics to Tables 1 and 2. The included strains are: 1, *Aequorivita marisscotiae* Ant34-E75; 2, *Aequorivita viscosa* 8-1b^T; 3, *Aequorivita lutea* q18^T; 4, *Aequorivita sinensis* S1-10^T; 5, *Aequorivita iocasae* KX20305^T; 6, *Aequorivita aquimaris* D-24^T; 7, *Aequorivita echinoideorum* CC-CZW007^T; 8, *Aequorivita vladivostokensis* KMM 3516^T; 9, *Aequorivita antarctica* SW49^T; 10, *Aequorivita lipolytica* Y10-2^T; 11, *Aequorivita soesokkakensis* RSSK-12^T. But some strains have incomplete information.

4. Before final acceptance, we also need the source data of your figures and graphs and especially all the processed RNAseq data including single values for all replicates as supplementary data set (e.g. an excel sheet).

Response: It has been provided as supplementary information in Supplementary Table 4 and Supplementary Table 8. Supplementary Table 4 contains a summary of all differentially expressed genes in strain Ant34-E75, while Supplementary Table 8 presents the relative mRNA levels obtained through quantitative real-time RT-PCR under different conditions.

Response to Reviewers

At the very beginning, I would like to show my gratitude for the reviewers and editor to give us the pertinent and useful comments. I have revised the manuscript according to most of the comments.

-Reviewer 1

Thank you very much for taking the time to review our manuscript.

1. Please update L107 78.06% to the new correct value according to Supp Table 1 (or simply 79%).

Response: It has been revised in line 107.

2. Please check/correct in the abstract - *Chlamydomonas reininatus* (*reinhardtii*?) and check manuscript for similar smaller mistakes.

Response: It has been revised in line 16. We have revisited the article to rectify these minor errors.